# Effects of Menstrual Cycle on the Accumulation of Human Papillomavirus-Infected Cells Exfoliated from the Cervix That Drift into the Vagina

**DOI:** 10.3390/microorganisms10040693

**Published:** 2022-03-23

**Authors:** Mitsuaki Okodo, Kaori Okayama, Koji Teruya, Kazumasa Tanabe, Chieko Ito, Yasuyoshi Ishii, Masahiko Fujii, Hirokazu Kimura, Mizue Oda

**Affiliations:** 1Department of Medical Technology, Faculty of Health Sciences, Kyorin University, 5-4-1 Shimorenjaku, Mitaka-shi, Tokyo 181-8621, Japan; tanakazu@ks.kyorin-u.ac.jp (K.T.); fujiim1951-1011@tbz.t-com.ne.jp (M.F.); 2Department of Health Science, Gunma Paz University Graduate School of Health Sciences, 1-7-1 Tonyamachi, Takasaki-shi, Gunma 370-0006, Japan; okayama@paz.ac.jp (K.O.); h-kimura@paz.ac.jp (H.K.); 3Department of Health and Welfare, Faculty of Health Sciences, Kyorin University, 5-4-1 Shimorenjaku, Mitaka-shi, Tokyo 181-8621, Japan; teruya@ks.kyorin-u.ac.jp; 4Department of Clinical Laboratory, Genki Plaza Medical Center for Health Care, 1-7-1 Jinbocho, Chiyoda-ku, Tokyo 101-0051, Japan; chie_nia.922@icloud.com (C.I.); y-ishii@genkiplaza.or.jp (Y.I.); 5Department of Gynecology, Genki Plaza Medical Center for Health Care, 1-105 Jinbocho, Chiyoda-ku, Tokyo 101-0051, Japan; m-oda@genkiplaza.or.jp

**Keywords:** human papillomavirus, cervical cancer screening, self-collection, menstrual cycle

## Abstract

Human papillomavirus (HPV) testing using self-collected vaginal specimens is the preferred choice to increase screening uptake. Although the HPV testing results of these samples depend on the cells that naturally exfoliate from the cervical lesion and drift into the vagina, the mechanism of when and how these exfoliated cells mix with the self-collected sample remains unclear. Hence, the study aimed to clarify the relationship between the vaginal drift of HPV-infected cells exfoliated from the cervix, and the menstrual cycle. A total of 180 scraped samples of the cervix and vagina were examined. The exfoliated cells were classified into two categories according to the HPV genotyping results of each sample: sufficient accumulation (same HPV types in cervical and vaginal samples) and insufficient accumulation (fewer HPV types in vaginal samples than in cervical samples, or HPV positivity in cervical samples and HPV negativity in vaginal samples). A moderately strong statistically significant association was observed between exfoliated cell accumulation and the menstrual cycle, and insufficient accumulation was statistically significantly increased at the early proliferative phases. Self-collection of vaginal samples at the early proliferation phase indicates insufficient sample quantities or lower viral load, thereby affecting HPV genotyping.

## 1. Introduction

Due to its greater sensitivity than cytology for detecting precancerous lesions, human papillomavirus (HPV) testing has gradually become the primary method for cervical cancer prevention programs [1,2]. Although cervical cancer screening helps decrease the incidence and mortality rate of cancer, it is still unlikely to resolve the problem of poor screening uptake [3]. In Japan, the participation rate in cervical cancer screening is quite low (approximately 40%) [3,4]. Numerous barriers prevent women from attending a screening; among them are embarrassment regarding the examination, fear of the test result, fear of pain, and lack of time [5]. Therefore, self-collection of vaginal samples for HPV testing has been proposed as an additional strategy to convince unwilling women [6,7,8]. Accumulating evidence reveals that the rate of detection of HPV and cervical intraepithelial neoplasia grade 2 or worse (CIN2+) in these self-collected specimens is comparable to that in physician-collected specimens [6,9,10,11,12,13].

The sources of HPV positive results in the vaginal sample are cells exfoliated from the female genitalia, including the cervix, and cells of unknown origin that were scraped where the self-sampling instrument contacted them [14]. If vaginal sampling fails to exfoliate a specific cervix, a false-positive result is possible because of other cells exfoliated from the vaginal canal, or a true-positive result because of cells that have naturally exfoliated and drifted away from the cervix. The fact that the HPV testing results of self-collected samples depend on the cells that have drifted away after natural exfoliation from various sites explains why their value is somewhat questionable. Nonetheless, the sensitivity of self-collected samples to CIN2+ is guaranteed to be equivalent to that of physician-collected samples as long as the self-collected samples contain even a small proportion of cells derived from cervical lesions. Hence, self-collection of vaginal samples for HPV testing can possibly be an alternative cervical cancer screening approach for women who are hesitant to undergo pelvic examination at a healthcare clinic. However, the mechanism of when and how exfoliated cells from cervical lesions mix with the self-collected sample remains poorly understood. Further investigation of the cells that exfoliate and drift away from the cervix is important to enhance the reliability of HPV testing using the self-collection approach, which can contribute to increasing the coverage of cervical cancer screening.

Therefore, using the HPV genotyping results of cervical and vaginal samples collected by physicians, this study aimed to clarify the relationship between HPV-infected cells that exfoliate from the cervix and drift into the vagina, and the menstrual cycle.

## 2. Materials and Methods

### 2.1. Patients and Specimen Collection

This study used two types of SurePath™ (Becton Dickinson and Company, Franklin Lakes, NJ, USA) liquid-based cytology (LBC) samples obtained from the cervix and vaginal canal of 324 patients. Prior to the study, these patients were referred for colposcopy because of abnormalities in cervical cytologic screening at the Tokyo Genki Plaza Health Medical Center from 2020 to 2021.

Immediately before colposcopic assessment, a vaginal speculum was inserted into the patient to expose the cervix, and then the endocervix and the portio vaginalis were scraped using an IM sampler instrument (Muto Pure Chemical, Tokyo, Japan), which includes a spatula and a brush, and eluted into a collection vial to collect the cervical sample. Next, with the vaginal speculum open, the bilateral parts from the vaginal fornix to the upper 1/2 of the vaginal canal was scraped and collected using a spatula for the vaginal sample. Written informed consent was obtained from the patients before sample collection. The Ethics Committee on Human Research of Kyorin University approved the study protocol. The protocol was implemented in accordance with the approved guidelines. Patients who had menopause, amenorrhea, or cervical and vaginal HPV negative by HPV genotyping (as described below) were excluded. We also excluded those with suspected vaginal preclinical HPV infection or vaginal intraepithelial neoplasia (VaIN) such as only low-risk (LR) HPV-type positive in cervical samples, HPV negative in cervical samples but HPV positive only in vaginal samples, or HPV positive in both samples but with more HPV types in vaginal samples. In particular, those who were only LR HPV-type positive in the cervical samples were excluded because LR types with suspected vaginal tropism [15] may have contaminated the cervix. Ultimately, 180 patients who had a high-risk (HR) or possibly-high-risk (pHR) HPV-type were evaluated (Figure 1). All patients were nonpregnant women of reproductive age, with a mean age of 36 (range: 22–53) years.

### 2.2. HPV Genotyping using LBC Samples

DNA from the cell pellet of the LBC samples was isolated using the hot sodium hydroxide method [16]. Cell pellets were lysed with 50 μL of alkaline lysis solution (25 mM NaOH and 0.2 mM ethylenediaminetetraacetic acid [EDTA]; pH, 12.0) for 30 min at 95 °C. Lysed cells were then neutralized with 0.04M Tris-HCl (pH 5.0), centrifuged at 13,200 rpm for 1 min, and directly used as the DNA template. The presence of human β-actin in cells was determined using polymerase chain reaction (PCR), which was an internal standard for genotyping. All HPV genotypes tested positive for human β-actin, demonstrating that we extracted DNA of amplifiable quality from the specimens. HPV genotyping was performed using uniplex E6/E7 PCR, a highly sensitive HPV-PCR method [17]. This method identified E6 or E7 genes of 39 mucosal HPV types, including 13 HR types (HPV16, 18, 31, 33, 35, 39, 45, 51, 52, 56, 58, 59, and 68), 11 pHR types (HPV26, 30, 34, 53, 66, 67, 69, 70, 73, 82, and 85), and 15 LR types (HPV6, 11, 40, 42, 44, 54, 55, 61, 62, 71, 74, 81, 84, 89, and 90), from as few as 100 viral copies, with no cross-reactivity across all the HPV genotypes.

### 2.3. Data Analysis

According to the HPV genotyping results of cervical and vaginal samples, we classified the HPV-infected cells that detached from the cervix and drifted into the vagina into two categories: sufficient and insufficient. HPV detected in the vagina is a possible result of contamination by exfoliated cells from the cervix [15]. Therefore, if the HPV types in the cervical and vaginal samples were the same, we classified these samples as sufficient accumulation of exfoliated cells in the vagina because it suggests that the HPV-infected cells came from the cervix and drifted into the vagina after natural exfoliation. Conversely, insufficient accumulation was considered if the HPV types in the vaginal samples were fewer than those in the cervical samples, or if HPV was positive in the cervical samples but negative in the vaginal samples, suggesting that the exfoliation of the cervical epithelium was suppressed, preventing the HPV-infected cells from drifting into the vagina.

Moreover, menstrual cycles were classified into five phases according to the date of the start of the last menstruation, days of menstrual cycle and bleeding period, and cyclic cytologic characteristics of the Pap smear: (1) menstrual phases (days 1–5; numerous red blood cells, leukocytes, and endometrial cells are present); (2) early proliferative phases (days 6–10; intermediate cells with degenerated cytoplasm are predominant, and endometrial cells and histiocytes are noted occasionally); (3) late proliferative phases (days 11–14; smear background is clear, and superficial cells with transparent flat acidophilic cytoplasm and pyknotic nuclei are predominant); (4) early secretory phases (days 15–21; superficial cells with basophil and folded cytoplasm are present, and the proportion of intermediate cells gradually increases); (5) late secretory phases (days 22–28; numerous *Döderlein bacilli*, leukocytes, and predominant intermediate cells with markedly folded or degraded cytoplasm are present).

Chi-square analysis with cross-tabulation frequencies was performed using Statistical Package for the Social Sciences version 25.0 (SPSS Inc., Chicago, IL, USA) to determine whether the accumulation of exfoliated cells was significantly different among the five menstrual phases. We also calculated adjusted residuals for each phase to aid in interpretation.

## 3. Results

Figure 1 illustrates how we classified the accumulated exfoliated cells in 180 patients according to the HPV genotyping results. All patients had no abnormal findings on colposcopic assessments of the vaginal fornix and vaginal canal. Table 1 enumerates the HPV types detected in the cervical and vaginal samples, the evaluation of the concordance of HPV types in each sample, and the menstrual cycle in each patient.

A total of 132 (73.3%) patients had the same HPV types detected in the cervical samples and vaginal samples, and were thereby classified as a sufficient accumulation of HPV-infected cells of cervical origin.

However, 26 (14.4%) patients had fewer HPV types detected in the vaginal sample than in the cervical sample. Those HPV types detected in cervical samples only were HR and pHR types, which accounted for 76.9% and 23.1%, respectively. Furthermore, 22 (12.2%) patients were HPV positive in the cervix but HPV negative in the vagina. The HPV genotypes detected in the cervical samples were HR type, pHR type, and a mixture of both in 86.4%, 9.1%, and 4.5%, respectively. Both cases were classified as insufficient accumulation of HPV-infected cells of cervical origin.

A chi-square test of independence was conducted between the accumulation of HPV-infected cells and the menstrual cycles. Results showed a moderately strong statistically significant association (χ2(4) = 10.642, *p* < 0.05) (Cramer’s V = 0.243). Table 2 presents a cross-tabulation of the accumulation of HPV-infected cells and the menstrual cycles. Each category consists of the observed frequencies, total percentage, expected frequencies, and adjusted residuals. Absolute adjusted residuals of >2.576 are highlighted in bold, and one category has >2.576 absolute adjusted residuals, indicating that insufficient accumulation was statistically significantly increased at the early proliferative phases (*p* < 0.01).

## 4. Discussion

Self-sampling HPV tests must be more consistent and reproducible with minimal false-negative results, considering that women who self-collect are distrustful of the results because they are uncertain whether they have collected the specimen correctly [18]. The present study demonstrated that the accumulation of HPV-infected cells that drifted from the cervical epithelium into the vagina is insufficient at the early proliferative phase. Therefore, self-collection of vaginal samples during the early proliferative phase of the menstrual cycle may lead to insufficient sample quantities and lower viral load, thereby affecting the HPV-DNA detection and genotyping.

The reason why insufficient exfoliated cells accumulate at the early proliferative phase may be that this phase is the period after the vaginal exfoliated cells have been washed away by menstrual blood; in addition, estrogen is gradually affecting the cervical epithelium, and the squamous epithelial layer is in the process of forming [19], resulting in a fewer cells available for exfoliation. The squamous epithelium that has differentiated to the superficial layer forms during the late proliferative phase, when estrogen secretion peaks. At this phase, the high estrogen effect increases the permeability and deformability of the cervical epithelium. This squamous epithelial change theoretically causes a decrease in epithelial cell connectivity [20], suggesting the start of natural exfoliation. At the beginning of the secretory phase, the progesterone effect causes deeper exfoliation of the superficial cells. In the final stage, intermediate cells are degraded and, during menstruation, the endometrium expulsion washes away cells on the cervical epithelium’s surface. Thus, if the physiological changes in the cervix caused by sex hormones are understood, we can infer that a constant number of cervical cells will naturally be exfoliated and accumulate in the vagina during most of the menstrual cycle. More than 70% of the women had the same genotypes in both cervical and vaginal samples; hence, natural cervical exfoliation and drift of the exfoliated cells into the vagina occur during most of the menstrual cycle. However, the timing of self-collection has some effect on the HPV test result.

Fairly et al. [21] investigated the effect of menstrual cycle on HPV detection rates and the size of tampons self-inserted into the vagina. The highest number of cells was collected in the middle of the menstrual cycle (probably the late proliferative to mid-secretory phase). In addition, the frequency of HPV detection was higher in this phase than in the early cycle, when the number of cells was the lowest. This result is consistent with the present study, in which insufficient accumulation of HPV-infected cells was observed during the early phase, when the amount of exfoliation was low. Studies that quantified the HPV copy numbers using daily self-collected vaginal samples showed consistency concerning daily HPV positivity and negativity in women, but HPV and cellular DNA copy numbers varied considerably between collection days [22]. Considering that they used a highly sensitive real-time PCR that can determine positive results for less than 10 copies, HPV results may not be consistent in PCR methods with limits of detection greater than 100 copies. Additionally, the copy numbers of viral and cellular DNA are almost positively correlated. Thus, the timing of self-collection may cause variations in the number of epithelial cells and HPV-infected cells in the sample, affecting the results of HPV testing.

However, Sherman et al. [23] found that the frequency of HPV positive results obtained by low-sensitivity Hybrid Capture 2 in physician-collected cervical samples was not related to the date of the last menstrual period. In another study, HPV typing results from similar samples were consistent regardless of the menstrual cycle. Given that the physician-collected cervical samples used in these studies mainly contain cells that have been scraped from cervical lesions, and only a portion of the cells have exfoliated naturally, any variation in the number of cells in the sample caused by the timing of collection may hardly affect the HPV testing results [24]. Therefore, the period of time when few cells have accumulated in the vagina should be considered because self-collection must target HPV-infected cells that have naturally exfoliated.

The strength of this study is that all cervical and vaginal samples were carefully collected by a single gynecologist sequentially on the same day of examination; thus, cross-contamination caused by collection was unlikely. Vaginal sampling also included the distal vagina to maximize the evaluation of the accumulation of naturally exfoliated cells drifting from the cervical epithelium. In the present study, we used a highly sensitive PCR method to detect wide-ranging HPV genotypes in both samples and subsequently detected periods with insufficient accumulation of exfoliated cells according to the results of HPV negativity or genotype reduction in the vaginal samples. Of note, all patients associated with vaginal preclinical HPV infection or VaIN were excluded. We believe that the result of undetected HR or pHR types in the vaginal samples indicates that these oncogenic genotypes have tropism [15] for the cervical epithelium and that the HPV-infected cells have not exfoliated from the cervix. This study also had some limitations. First, when the same HPV genotypes were detected in the cervical and vaginal samples, we determined that HPV-infected cells naturally exfoliated from the cervix and reached the vagina. Considering that more than 70% of the patients had the same HPV-type detected in both samples, having the same type that infected each epithelium seems unlikely. However, these two anatomical locations are attached close to each other. Furthermore, although no vaginal lesions were found by colposcopy, the HR type HPV-infected cells present in the vagina were possibly sampled after adhering to the cervix and were then classified as exfoliated cell accumulation. Although most samples with discordant cervical and vaginal HPV test results suggesting insufficient cell accumulation were collected in the early proliferative phase, this finding may have a limited value. Second, our statistics and inferences were restricted because of the limited number of samples included in this study. In addition, this study only speculated about the impact of self-collected samples on HPV testing according to cell exfoliation results obtained in a small population of women who visited for a detailed examination. Further validation studies using self-collected samples in a larger study population are needed to clarify the impact of the timing of each self-collection during the menstrual cycle on HPV testing.

## 5. Conclusions

The HPV genotyping results of the cervical and vaginal samples revealed that the accumulation of exfoliated cells in the vagina is insufficient at the early proliferative phase of the menstrual cycle. This finding provides new insights for standardization of the optimal timing for self-collection and contributes to the growing literature on the effect on HPV test results in self-collected samples.

## Figures and Tables

**Figure 1 microorganisms-10-00693-f001:**
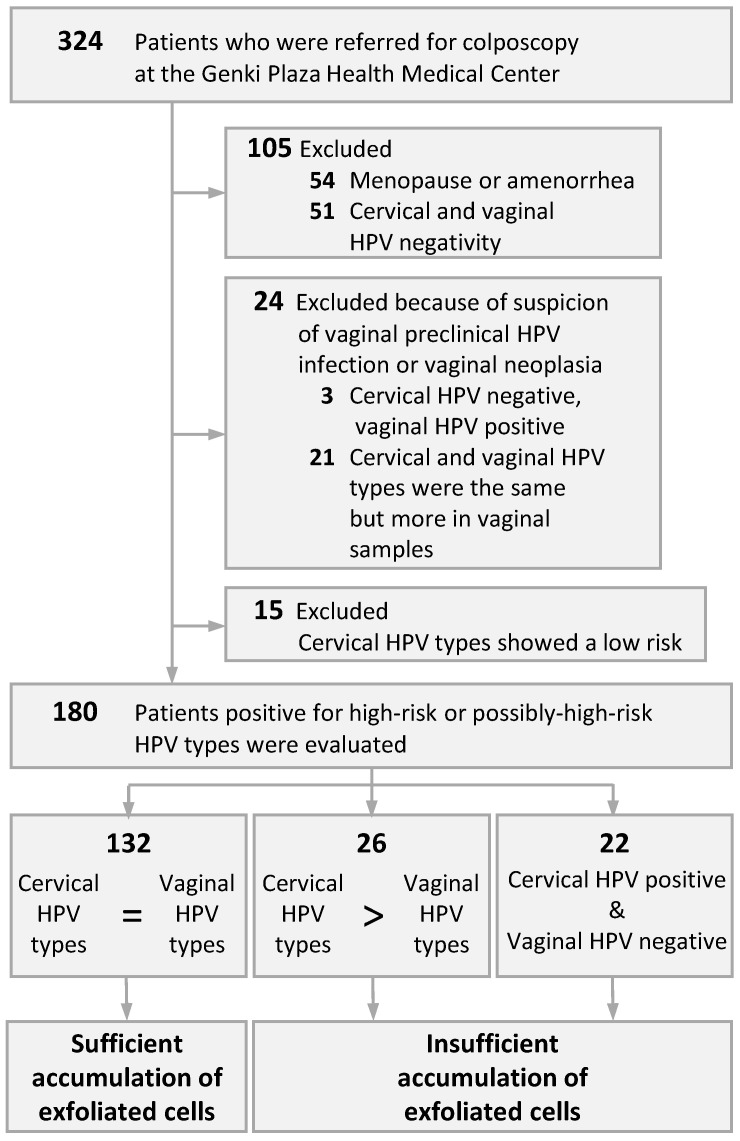
Stratification of patients by the accumulation of exfoliated cells during the menstrual cycle. HPV, human papillomavirus.

**Table 1 microorganisms-10-00693-t001:** Accumulation of exfoliated cells and menstrual cycle in 180 patients.

Case	Cervical HPV Types	Vaginal HPV Types	Evaluation	Cycles	Case	Cervical HPV Types	Vaginal HPV Types	Evaluation	Cycles
1	16,52,58,66	16,52,58,66	C = V	L-Pro	91	66	66	C = V	L-Sec
2	52,6b	52,6b	C = V	L-Sec	92	52,42,74	52,42,74	C = V	Mens
3	51,58	51,58	C = V	E-Pro	93	52	52	C = V	Mens
4	**31**,81,82	81,82	C > V	E-Pro	94	**31**,58,74	58,74	C > V	E-Pro
5	18	18	C = V	E-Sec	95	16	16	C = V	E-Sec
6	**67**	Negative	N	L-Sec	96	39,58,53	39,58,53	C = V	E-Sec
7	**16**	Negative	N	L-Sec	97	51	51	C = V	L-Sec
8	52	52	C = V	L-Pro	98	16	16	C = V	L-Pro
9	82,90	82,90	C = V	E-Pro	99	59,90	59,90	C = V	E-Sec
10	33,52	33,52	C = V	L-Pro	100	**52**	Negative	N	L-Sec
11	52	52	C = V	L-Pro	101	31,52,69,74	31,52,69,74	C = V	E-Pro
12	52,66	52,66	C = V	L-Sec	102	52,6b	52,6b	C = V	L-Sec
13	56	56	C = V	E-Sec	103	16	16	C = V	L-Sec
14	16,45,**52**,40	16,45,40	C > V	E-Pro	104	52	52	C = V	L-Pro
15	58	58	C = V	Mens	105	56	56	C = V	E-Sec
16	16	16	C = V	L-Pro	106	66	66	C = V	L-Sec
17	**56**,62	62	C > V	L-Sec	107	31,**52**,58,70	31,58,70	C > V	L-Pro
18	52	52	C = V	E-Sec	108	**52**	Negative	N	E-Pro
19	39,58,53	39,58,53	C = V	E-Sec	109	51,58,**82**	51,58	C > V	Mens
20	16,66	16,66	C = V	L-Sec	110	58	58	C = V	L-Sec
21	33	33	C = V	L-Sec	111	16,51,52,56	16,51,52,56	C = V	L-Sec
22	52	52	C = V	E-Sec	112	58	58	C = V	L-Pro
23	**52**,53,74	53,74	C > V	L-Sec	113	31,71	31,71	C = V	E-Sec
24	52,84	52,84	C = V	E-Sec	114	51,82	51,82	C = V	L-Sec
25	**56**	Negative	N	L-Sec	115	51,53,54	51,53,54	C = V	E-Pro
26	**51**,58	58	C > V	E-Sec	116	39,59,40	39,59,40	C = V	L-Sec
27	52,**58**	52	C > V	L-Sec	117	56,66,74	56,66,74	C = V	L-Pro
28	51,58,82	51,58,82	C = V	E-Sec	118	53,**66**	53	C > V	L-Pro
29	56,61,74	56,61,74	C = V	E-Sec	119	66	66	C = V	Mens
30	39,71	39,71	C = V	L-Sec	120	58	58	C = V	E-Sec
31	16	16	C = V	L-Pro	121	16	16	C = V	L-Pro
32	52,54,62,70,90	52,54,62,70,90	C = V	L-Sec	122	58	58	C = V	L-Pro
33	**39**,58	58	C > V	E-Sec	123	16	16	C = V	L-Pro
34	53,81,90	53,81,90	C = V	L-Sec	124	58	58	C = V	E-Sec
35	**51**	Negative	N	E-Pro	125	**52**	Negative	N	L-Sec
36	**52**	Negative	N	E-Pro	126	58	58	C = V	E-Pro
37	**16**	Negative	N	L-Pro	127	39,51,53,42	39,51,53,42	C = V	E-Sec
38	52,56	52,56	C = V	E-Pro	128	68	68	C = V	E-Sec
39	31	31	C = V	L-Sec	129	**31**	Negative	N	E-Sec
40	51,71	51,71	C = V	E-Pro	130	51,71	51,71	C = V	E-Pro
41	59,74	59,74	C = V	E-Pro	131	34	34	C = V	L-Sec
42	52	52	C = V	E-Sec	132	66,62,81	66,62,81	C = V	L-Sec
43	51,82	51,82	C = V	L-Sec	133	82,90	82,90	C = V	Mens
44	**51**	Negative	N	E-Sec	134	53	53	C = V	L-Sec
45	**31**,74	74	C > V	E-Sec	135	52	52	C = V	L-Sec
46	52	52	C = V	E-Sec	136	31	31	C = V	L-Sec
47	16	16	C = V	L-Pro	137	62,81,90	62,81,90	C = V	E-Sec
48	52	52	C = V	L-Sec	138	**18**	Negative	N	L-Sec
49	**16**,52,56	52,56	C > V	E-Sec	139	51	51	C = V	E-Sec
50	52,**58**	52	C > V	E-Sec	140	52	52	C = V	L-Sec
51	58,67	58,67	C = V	E-Sec	141	58	58	C = V	E-Pro
52	16,31,51,52,58,82,54,70	16,31,51,52,58, 82,54,70	C = V	L-Pro	142	51,82,62	51,82,62	C = V	E-Sec
53	52,6b	52,6b	C = V	L-Pro	143	**51**,74	74	C > V	L-Pro
54	58	58	C = V	E-Sec	144	53,40,62,81	53,40,62,81	C = V	L-Sec
55	53	53	C = V	E-Sec	145	16,39	16,39	C = V	E-Sec
56	52,40,81	52,40,81	C = V	Mens	146	67	Negative	N	E-Sec
57	52	52	C = V	L-Sec	147	52	52	C = V	E-Sec
58	56,66	56,66	C = V	E-Sec	148	**56**	Negative	N	E-Pro
59	52,62	52,62	C = V	Mens	149	51,82	51,82	C = V	E-Sec
60	51,53,54	51,53,54	C = V	E-Pro	150	31	31	C = V	L-Pro
61	**56**,34	34	C > V	L-Pro	151	**51**	Negative	N	L-Sec
62	**16**	Negative	N	L-Sec	152	31,**51,52**,58	31,58	C > V	E-Pro
63	16,66	16,66	C = V	E-Sec	153	53,74	53,74	C = V	L-Sec
64	52,58	52,58	C = V	E-Pro	154	**66**,40	40	C > V	E-Sec
65	**52**	Negative	N	E-Pro	155	16,66	16,66	C = V	L-Sec
66	**51,82**	Negative	N	Mens	156	31	31	C = V	L-Pro
67	82	82	C = V	E-Sec	157	58	58	C = V	L-Pro
68	68	68	C = V	Mens	158	31,90	31,90	C = V	E-Sec
69	45,59,53,62,66,67,81	45,59,53,62,66, 67,81	C = V	L-Sec	159	58	58	C = V	E-Sec
70	51	51	C = V	L-Pro	160	31,33,53,68	31,33,53,68	C = V	E-Pro
71	53	53	C = V	L-Pro	161	31,68,67	31,68,67	C = V	E-Sec
72	59,74	59,74	C = V	E-Sec	162	56	56	C = V	L-Sec
73	51,**82**	51	C > V	L-Pro	163	52	52	C = V	L-Pro
74	**52**	Negative	N	E-Pro	164	53	53	C = V	L-Pro
75	33,**52**	33	C > V	E-Pro	165	**26**,90	90	C > V	L-Sec
76	52	52	C = V	E-Sec	166	16	16	C = V	E-Sec
77	51,**58**	51	C > V	L-Pro	167	52,67,74,89	52,67,74,89	C = V	L-Pro
78	33	33	C = V	L-Sec	168	66,30	66,30	C = V	E-Sec
79	31,90	31,90	C = V	L-Sec	169	16	16	C = V	E-Pro
80	39,59,40	39,59,40	C = V	E-Sec	170	**52**,59	59	C > V	E-Pro
81	58	58	C = V	E-Sec	171	16,51,52,61	16,51,52,61	C = V	E-Sec
82	**51**,61,74	61,74	C > V	E-Sec	172	16	16	C = V	E-Pro
83	66	66	C = V	E-Sec	173	52,82	52,82	C = V	Mens
84	**66**,40	40	C > V	E-Pro	174	**33,52**	Negative	N	E-Pro
85	**56**	Negative	N	L-Sec	175	58	58	C = V	L-Pro
86	59	59	C = V	E-Sec	176	51	51	C = V	E-Pro
87	31,52,69,42,55,74	31,52,69,42,55, 74	C = V	E-Sec	177	82	82	C = V	E-Sec
88	52,67,74	52,67,74	C = V	L-Sec	178	18	18	C = V	L-Pro
89	16,52	16,52	C = V	L-Pro	179	66	66	C = V	L-Sec
90	**52**	Negative	N	E-Pro	180	**18**,31,52,**59**,67,55,74,90	31,52,67,55,74,90	C > V	L-Sec

The HPV-type column for each case is shown in the order of the high-risk (HR), possibly-high-risk (pHR), and low-risk (LR) types. Different HPV types in cervical and vaginal samples are highlighted in bold. C = V, HPV types in the cervix and vaginal samples are the same; N, HPV is positive in the cervical samples but negative in the vaginal samples; C > V, the number of HPV types in the vaginal samples was reduced more than in the cervical samples; HPV, human papillomavirus; Mens, menstrual phase; E-Pro, early proliferative phase; L-Pro, late proliferative phase; E-Sec, early secretory phases; L-Sec, late secretory phases.

**Table 2 microorganisms-10-00693-t002:** Cross-tabulation of the menstrual cycle phases and the accumulation of exfoliated cells.

		Accumulation of Exfoliated Cells	
		Sufficient	Insufficient
Menstrual phases	Frequencies	9	2
	Total percentage	82%	18%
	Expected frequencies	8.1	2.9
	Adjusted residual	0.7	−0.7
Early proliferative phase	Frequencies	16	15
	Total percentage	52%	48%
	Expected frequencies	22.7	8.3
	Adjusted residual	**−3.0 ***	**3.0 ***
Late proliferative phase	Frequencies	27	7
	Total percentage	79%	21%
	Expected frequencies	24.9	9.1
	Adjusted residual	0.9	−0.9
Early secretory phases	Frequencies	45	10
	Total percentage	82%	18%
	Expected frequencies	40.3	14.7
	Adjusted residual	1.7	−1.7
Late secretory phases	Frequencies	35	14
	Total percentage	71%	29%
	Expected frequencies	35.9	13.1
	Adjusted residual	−0.4	0.4

Absolute adjusted residuals of > 2.576 are highlighted in bold. * *p* < 0.01.

## Data Availability

The data and material that support the findings of this study are available from the corresponding author upon reasonable request.

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
