# Peer review of "Effects of Menstrual Cycle on the Accumulation of Human Papillomavirus-Infected Cells Exfoliated from the Cervix That Drift into the Vagina"

_microorganisms, 2022, doi:10.3390/microorganisms10040693_

Round 1

Reviewer 1 Report

file adnext

Reviewer 2 Report

Okodo et al. have studied the relationship between the menstrual cycle and the vaginal drift of HPV-infected cells exfoliated from the cervix. Samples of the cervix and vagina scraped were examined in 180 women. A moderately strong statistically significant association was observed between exfoliated cell accumulation and the menstrual cycle, insufficient accumulation was statistically significantly increased at the early proliferative phases. In conclusion, Self-collection of vaginal and urine samples at the early proliferation phase suggests insufficient sample quantities or lower viral load, thereby affecting HPV genotyping.

Comments

  1. Cervical cancer can be prevented through screening by identifying and treating the precancerous lesions. HPV testing in primary screening is a more sensitive method of detecting high-grade cervical lesions than Pap smears The purpose of HPV testing is not to find all women with HPV infections, but women with pre-stages of cervical cancer. All the 180 women included in the study were referred to colposcopy. Still, there is no results from colpocopies and biopsies included in the manuscript.
  2. Most clinically validated HPV-test have a high cut-off to balance benefits and harms. For eksample Hybrid Capture II (HC2) have a cut-of 5.000 viral copies in the samle (RLU/CO value of ≥1.0). In this study, HPV genotyping was performed using uniplex E6/E7 PCR, a highly sensitive HPV-PCR method which can detect as few as 100 viral copies. This will increase sensitivity for CIN2+, but decrease specificity.
  3. In the study, the HPV-test included 39 mucosal HPV types, including 13 HR types, 11 pHR types, and 15 LR types. Usually the analysis should be restricted to the 13 hrHPV-types used in clinically validated HPV-tests.
  4. Both self-collected vaginal specimens and urine specimens are discussed in the manuscript, but only vaginal specimens are included in the study. In my experience, vaginal specimens are more reliable than urine samples for HPV-testing in cervical cancer prevention. You can not draw any conclusions of results of urine samples when studying cervical anc vaginal samples only.

Minor revisions

Line 49-50, "However, the value of HPV testing using self-collected specimens remains controversial [13]."

First of all, the reference 13 is about HPV-testing in urine, not self-collected specimens from the vagina.

Second, I disagree. HPV self-sampling has been widely supported by the scientific community following a strong body of literature on the subject. Selfcollection of cervical samples is reported to be highly acceptable and preferred by most women, being a promising approach to enhance women’s participation in regular screening for cervical cancer prevention (Sultana 2015). It offers significant benefits over conventional sampling in terms of
cost, coverage and convenience for patients. Self-sampling reaches high-risk groups who currently have limited access to national health system screening for personal and practical reasons (Mariño, 2015). Numerous studies comparing selfcollected and clinician-collected samples for HPV detection show good concordance when clinically validated PCR-based methods are used (Petignat 2007, Schmeinkt 2011). In a meta-analysis by Arbyn et al., self-sampled HPV tests based on PCR for the detection of CIN2+ were shown not to have statistically different sensitivity or specificity compared with cliniciansampled tests (Arbyn 2014).

In the Netherlands, women aged 30–60 years invited for cervical screening can choose between sampling at the clinician's office (Cervex Brush) or self-sampling at home (Evalyn Brush). Between January 2017 and March 2018 a total of 30,808 women had a self-collected and 456,207 had a clinician-collected sample. The relative sensitivity for detecting CIN3+ was 0.94 (0.90–0.97) for self-collection versus clinician-collection and the relative specificity was 1.02 (1.02–1.02). The clinical accuracy of hrHPV testing on a self-collected sample for detection of CIN3+ is high and supports its use as a primary screening test for all invited women (Inturrisi 2021). 

In a study from Mexico with 505 women, 96.8% of the participants reported they felt confident carrying out the self-collection themselves, and 88.8% reported no discomfort at all performing the procedure (Aranda 2021),

Table 1 is large, very complex and should be moved to "Supplemental". In the legend of the table it states to include 180 women, but there are 324 cases listed. Please explain. In Figure 1 you started with 324 women, but included only 180 women in the final analysis.

Line 236-237, "the sensitivity of HPV testing using self-collected samples is reportedly 10% lower than that using physician-collected samples"

add "In the Nederlands, the relative sensitivity for detecting CIN3+ was 0.94 (0.90–0.97) for self-collection versus clinician-collection and the relative specificity was 1.02 (1.02–1.02) (Inturrisi 2021)."

References

Sultana F, Mullins R, English DR, Simpson JA, Drennan KT, Heley S, et al.
Women's experience with home-based self-sampling for human papillomavirus testing. BMC Cancer. 2015;15(1):849. https://doi.org/10.1186/
s12885-015-1804-x.

Mariño H, Serra E, Gutiérrez A. Self-sampling is as much effective as
gynecologist samples for HPV detection. Medicina Balear. 2015;30:16–20.

Petignat P, Faltin DL, Bruchim I, Tramer MR, Franco EL, Coutlee F. Are selfcollected samples comparable to physician-collected cervical specimens for human papillomavirus DNA testing? A systematic review and meta-analysis. Gynecol Oncol. 2007;105(2):530–5. https://doi.org/10.1016/j.ygyno.2007.01.023

Schmeink CE, Bekkers RL, Massuger LF, Melchers WJ. The potential role of self-sampling for high-risk human papillomavirus detection in cervical
cancer screening. Rev Med Virol. 2011;21(3):139–53. https://doi.org/10.1002/
rmv.686.

Arbyn M, Verdoodt F, Snijders PJ, Verhoef VM, Suonio E, Dillner L, et al.
Accuracy of human papillomavirus testing on self-collected versus cliniciancollected samples: a meta-analysis. Lancet Oncol. 2014;15(2):172–83. https://doi.org/10.1016/S1470-2045(13)70570-9.

Inturrisi F, Aitken CA, Melchers WJG, van den Brule AJC, Molijn A, Hinrichs JWJ, Niesters HGM, Siebers AG, Schuurman R, Heideman DAM, de Kok IMCM, Bekkers RLM, van Kemenade FJ, Berkhof J. Clinical performance of high-risk HPV testing on self-samples versus clinician samples in routine primary HPV screening in the Netherlands: An observational study. Lancet Reg Health Eur. 2021 Nov 9;11:100235. doi: 10.1016/j.lanepe.2021.100235. PMID: 34918001; PMCID: PMC8642706.

https://pubmed.ncbi.nlm.nih.gov/34918001/

Aranda Flores CE, Gomez Gutierrez G, Ortiz Leon JM, et al. Self-collected versus clinician-collected cervical samples for the detection of HPV infections by 14-type DNA and 7-type mRNA tests. BMC Infect Dis. 2021 May 31;21(1):504. doi: 10.1186/s12879-021-06189-2. 

https://pubmed.ncbi.nlm.nih.gov/34058992/

Round 2

Reviewer 1 Report

I accept the present form

Reviewer 2 Report

All comments have been addressed. The response to the comments is adequate and improved the manuscript.